# Deciphering the Antifibrotic Property of Metformin

**DOI:** 10.3390/cells11244090

**Published:** 2022-12-16

**Authors:** Axelle Septembre-Malaterre, Chailas Boina, Audrey Douanier, Philippe Gasque

**Affiliations:** 1Unité de Recherche, EPI ‘Etudes en Pharmaco-Immunologie’, Université de la Réunion, Allée des Topazes, CS11021, 97400 Saint Denis, France; 2Laboratoire D’immunologie Clinique et Expérimentale de la Zone de L’océan Indien (LICE-OI), CHU La Réunion Site Félix Guyon Allée des Topazes, CS11021, 97400 Saint Denis, France

**Keywords:** mesenchymal stem cells, myofibroblast, transdifferentiation, inflammation, miRs, oxidative stress, fibrosis, metformin

## Abstract

Fibrosis is a chronic progressive and incurable disease leading to organ dysfunction. It is characterized by the accumulation of extracellular matrix proteins produced by mesenchymal stem cells (MSCs) differentiating into myofibroblasts. Given the complexity of its pathophysiology, the search for effective treatments for fibrosis is of paramount importance. Metformin, a structural dimethyl analog of the galegine guanide extracted from the “French Lilac” (Fabaceae *Galega officinalis*), is the most widely used antidiabetic drug, recently recognized for its antifibrotic effects through ill-characterized mechanisms. The in vitro model of TGF-β1-induced fibrosis in human primary pulmonary mesenchymal stem cells (HPMSCs), identified as CD248+ and CD90+ cells, was used to study the effects of metformin extracts. These effects were tested on the expression of canonical MSC differentiation markers, immune/inflammatory factors and antioxidative stress molecules using qRT-PCR (mRNA, miRNA), immunofluorescence and ELISA experiments. Interestingly, metformin is able to reduce/modulate the expression of different actors involved in fibrosis. Indeed, TGF-β1 effects were markedly attenuated by metformin, as evidenced by reduced expression of three collagen types and Acta2 mRNAs. Furthermore, metformin attenuated the effects of TGF-β1 on the expression of PDGF, VEGF, erythropoietin, calcitonin and profibrotic miRs, possibly by controlling the expression of several key TGF/Smad factors. The expression of four major fibrogenic MMPs was also reduced by metformin treatment. In addition, metformin controlled MSC differentiation into lipofibroblasts and osteoblasts and had the ability to restore redox balance via the Nox4/Nrf2, AMP and Pi3K pathways. Overall, these results show that metformin is a candidate molecule for antifibrotic effect and/or aiming to combat the development of chronic inflammatory diseases worldwide.

## 1. Introduction

Pathological fibrosis is due to dysregulated repair responses following repeated injuries, which result in loss of organ function and tissue damage [1]. These injuries can be caused by respirable toxins (cigarette smoking), host cellular debris (alarmins, e.g., HMGB1) or viral infections [2,3]. The pathophysiology of pulmonary fibrosis is particularly complex because it involves multiple cellular actors whose roles evolve over time. As a result, the cellular and molecular mechanisms responsible for the fibrosis process are not completely known and are still frequently based on hypotheses derived from experimental models of fibrosis developed in rodents.

The hallmark lesions of idiopathic pulmonary fibrosis are the fibroblast foci. These sites feature vigorous proliferation of fibroblasts and exuberant deposition of an excessive extracellular matrix (ECM). These activities could be explained by potentially impaired matrix metalloproteases (MMPs) and failure to control ECM depots. Fibroblast foci are typical of alveolar epithelial cell injury, with endoluminal plasma exudation and collapse of the distal air space [4]. Stressed epithelial cells will release various growth factors and chemokines (platelet-derived growth factor (PDGF); chemokine ligand 12 (CXCL12), also known as SDF-1), which will create a favorable microenvironment for the activation, proliferation and differentiation of the resident pulmonary fibroblasts and circulating fibroblasts (fibrocytes) into myofibroblasts at the site of injury [5,6,7]. Several line-tracing experiments using transgenic mice (e.g., fluorescent protein expression under Gli1 or myelin P zero) have revealed that these (myo)fibroblasts are derived from perivascular mesenchymal stem/stromal cells (pMSCs) originating from the neural crest embryonic tissue [8,9]. These pMSCs are known to express markers such as CD90 (GPI-anchored Thy1) and CD248 (C-type lectin endosialin), particularly when they are activated. In light of their neural crest origin, MSCs (as well as myofibroblasts) can transdifferentiate (or dedifferentiate) into osteoblasts, lipofibroblasts and Schwann/neuronal cell phenotypes, indicated by markers such as (RUNX2, SOST) (PPAR-γ, PLIN2) and (myelin P zero, GFAP), respectively [8]. These pMSCs also have important hormonal activities and contribute, for instance, to the production of erythropoietin (EPO) and calcitonin [10,11]. Myofibroblasts are characterized by the expression of α-smooth muscle actin (α-SMA), also called the coding gene actin alpha 2 (Acta2). Alveolar epithelial cells are also the main sources of profibrotic factors, such as transforming growth factor-beta 1 (TGF-β1), platelet-derived growth factor (PDGF), tumor necrosis factor-α (TNF-α) and endothelin-1 [12,13]. Myofibroblasts synthesize an abnormally rigid ECM composed of an excessive amount of collagen (Col) and fibronectins [14,15,16].

These processes are accompanied by an inflammatory response, mainly controlled by cytokines such as interleukin-1 beta (IL-1β) and tumor necrosis factor-alpha (TNF-α), which contributes to macrophage activation and polarization [17]. Perivascular MSCs, as well as leukocytes, are mobilized and recruited by chemotaxis. The recruitment to the site of injury is regulated by chemokines such as CCL2 (MCP-1), CXCL8 (IL-8) and CCL5 (RANTES). The inflammatory environment subsequently stimulates MSCs to secrete paracrine growth factors (e.g., vascular endothelial growth factor (VEGF)) that collectively play a role in supporting the regenerative process and controlling fibrosis in injured tissue [18].

Numerous studies have shown a correlation between the deregulation of the expression of some miRNAs and the development of fibrosis [19]. The mapping of genes coding for miRNAs is at an early stage, but among the best characterized, we can cite miR-21-23a-29-30c-199-449 and let-7. They notably illustrate how miRNAs can act as pro- or antifibrotic agents [20,21,22]. Liu and colleagues demonstrated, by using in situ hybridization, a marked increase in miR-21 expression in the lungs of mice with pulmonary fibrosis and in the lungs of patients with idiopathic pulmonary fibrosis, as well as colocalization of miR-21 with myofibroblasts. Furthermore, bleomycin-induced pulmonary fibrosis was reduced by administration of antagomiRs directed against miR-21 in mouse models (C57BL/6). miR-21 functions in a positive feedback loop system under the action of TGF-β1 [23]. miR-199a-5p is overexpressed during the fibrosis process. It induces activation of pulmonary fibroblasts and their differentiation into myofibroblasts [24]. Caveolin-1 (CAV-1) has been identified as the target of miR-199. TGF-β1 induces a decrease in the expression of CAV-1 [25]. A study conducted by Cardenas et al. showed the role of miR-199a-5p in the TGF-β signaling pathway. Oligonucleotides were used to inhibit miR-199a-5p, specifically by blocking the binding to its target CAV-1 [26]. miR-214 is also overexpressed in response to TGF-β1 during the fibrogenesis process [27]. Conversely, miR-29 exerts an antifibrotic effect. Indeed, increased expression of miR-29 in fibroblasts decreases the production of type I and type III collagen, suggesting a post-transcriptional effect of miR-29. The antifibrotic effects of miR-29 were observed in lung, kidney and myocardial fibrosis [28].

Autophagy has also been described as being involved in fibrotic disorders [29]. Indeed, a decrease in the expression of the autophagic marker LC3 in lung tissues of patients with pulmonary fibrosis indicates that autophagy activity may be impaired [30]. Regulation of autophagy can be mediated by several intracellular pathways, such as the PI3K/Akt, AMPK and MAPK signaling pathways [31]. These pathways are known to affect the pathway downstream of the TGF-β receptor and involve different SMAD molecules.

Several mediators are involved in pulmonary fibrosis and seem to be implicated at different stages of the disease. Many studies have shown the importance of oxidative stress and its participation in the pathophysiology of pulmonary fibrosis [32]. Reactive oxygen species (ROS), in particular H_2_O_2_, are modulators of cellular differentiation. They are implicated in the differentiation of resident fibroblasts into pulmonary myofibroblasts, which can contribute to the development of fibrosis [33,34]. Membrane Nox4/NADPH oxidase expressed by lung fibroblasts is thought to be responsible for excessive ROS production and is a potential therapeutic target. The transcription factor Nrf2 also plays a role in modulating the cellular phenotype during fibrotic processes [35].

The incidence of pulmonary fibrosis is gradually increasing. Despite considerable progress in recent years, there is no effective treatment and the prognosis of this pathology is extremely poor [36].

Only two drugs, pirfenidone and nintedanib, are approved as effective therapies in the clinic, and neither of these agents has been able to stop the progression of pulmonary fibrosis [37,38,39]. Pirfenidone is a pyridone molecule that has been reported to block in vitro growth factor-stimulated collagen synthesis, extracellular matrix secretion and fibroblast proliferation [40]. Nintedanib is a tyrosine kinase inhibitor that targets growth factor pathways, including those downstream of VEGF receptors 1, 2 and 3; fibroblast growth factor (FGF) receptors 1, 2 and 3; and PDGF receptors.

The identification and discovery of new agents for pulmonary fibrosis remain urgent. Metformin is a synthetic biguanide analogous to galegine extracted from the French Lilac Fabaceae tree, both known for their antidiabetic activities. Critically, several studies have also reported that metformin improves tumor progression, inflammatory diseases and tissue fibrosis [41,42]. Several reports have claimed that metformin can reverse the bleomycin-induced mouse lung fibrosis pattern [41,43]. However, it is not known whether metformin attenuates TGF-β1-induced pulmonary fibrosis via inhibition of HPMSC transdifferentiation. This mechanism in this cell model has not yet been clarified. Further studies on its mechanism of action and its clinical efficacy are highly required.

In the present study, we sought to investigate whether metformin (Met) could have antifibrotic effects. We used recombinant TGF-β1 to induce a profibrotic differentiating phenotype of primary human pulmonary MSCs, whereas PDGF BB-treated MSCs were used to ascertain the effects on MSC activation.

## 2. Materials and Methods

### 2.1. Cells and Reagents

HPMSCs were obtained from ScienCell research laboratory (ScienCell, Carlsbad, CA, USA, Cat. No.: 7540) and grown in Modified Eagle’s Medium (MEM, PAN Biotech P0408500, Aidenbach, Germany) supplemented with 10% heat-inactivated fetal bovine serum (FBS; PAN Biotech, 3302 P290907), 2 mM of L-glutamine (Biochrom AG, Berlin, Germany, K0282), 0.1 mg/mL penicillin–streptomycin (PAN Biotech, P0607100), 1 mM of sodium pyruvate (PAN Biotech, P0443100) and 0.5 µg/mL of amphotericin B (PAN Biotech, P0601001). Recombinant human PDGF and TGF-β1 were purchased from PeproTech (Cranbury, NJ, USA). Metformin was purchased from Sigma (St. Louis, MO, USA).

### 2.2. Cell Culture

HPMSCs were placed in a 96-, 24- or 6-well plate and maintained at 37 °C in a humid atmosphere with 5% CO_2_. The medium was replaced twice a week. Cells were treated with TGF-β1 or PDGF at 10 ng/mL in association or not with metformin (Met) 5 mM for 72 h at 37 °C in a humid atmosphere with 5% CO_2_. These cells were previously deprived of SVF for 6 h.

### 2.3. Cytotoxicity Assay

Cytotoxicity assay was performed by measuring the release of lactate dehydrogenase (LDH) from damaged cells using a colorimetric-based kit (Ref.: G1781, CytoTox 96^®^ Non-Radioactive Cytotoxicity Assay, Promega, Madison, WI, USA). Cells were grown as previously described and treated with TGF-β1 or PDGF at 10 ng/mL in association or not with Met 5 mM for 72 h. Culture medium was collected and cells were lysed following the manufacturer’s instructions. Released LDH in culture medium after treatments was compared to the maximum LDH release (intracellular LDH induced by addition of Triton 1%). Cytotoxicity was expressed relative to maximum LDH release with the formula: % cytotoxicity = 100 × experimental LDH release/maximum LDH release control (CTRL), which refers to LDH release from untreated cells.

For cell mitochondrial metabolic activity measurements, 3-(4-5-dimethylthiazol-2-yl)-2,5-diphenyltetrazolium bromide (MTT) assay was performed according to the method of Mosmann [44]. Cells were plated in 96-well plates at a density of 4 × 103 cells/well. After 24 h, the culture medium was removed, and cells were exposed to TGF-β1 or PDGF at 10 ng/mL in association or not with Met 5 mM for 72 h. Five hours before the end of the experiment, 20 µL of sterile-filtered MTT solution (5 mg/mL) (Sigma-Aldrich, Darmstadt, Germany), prepared in phosphate-buffered saline (PBS), was added to each well, and the plate was incubated at 37 °C. Then, the unreacted dye was removed by centrifugation, the insoluble formazan crystals were dissolved in 200 µL/well dimethyl sulfoxide and the absorbance was measured at 560 nm (Biotek Cytation 5 imaging reader).

### 2.4. Immunofluorescence Staining and Microscopy to Assess Cell Marker Expression

Cells were grown on glass coverslips in a 24-well plate, as previously described, then treated with TGF-β1 or PDGF at 10 ng/mL in association or not with Met 5 mM for 72 h. Coverslips were then fixed and permeabilized with cold 99% ethanol at room temperature for 10 min. Cells were incubated overnight with primary antibodies against either human collagen 1 (Col1) (Mouse Monoclonal Clone M38, DSHB, Iowa City, IA, USA). Alexa Fluor^®^ 488-conjugated donkey anti-mouse (Invitrogen, Thermo Fisher Scientific, Waltham, MA, USA) was used as a secondary antibody. The nucleus was revealed by staining with nuclear fluorochrome 4′,6-diamidino-2-phenylindole (DAPI). The coverslips were mounted with VECTASHIE:LD (Vector Labs, Newark, CA, USA), and fluorescence was observed using a Nikon Eclipse E2000-U microscope. Images were captured and processed using a Hamamatsu ORCA-ER camera and the imaging software NIS-Element AR (Nikon, Tokyo, Japan). Magnification was ×60.

### 2.5. Histo-ELISA Assay

Culture media collected from cells treated with TGF-β1 or PDGF at 10 ng/mL in association or not with Met 5 mM for 72 h were analyzed by histo-ELISA, as previously described. CXCL-12 concentrations in HPMSC supernatants were measured using commercially available ELISA kits for CXCL-12 (PeproTech, Cat. No.: 900-K92), according to the manufacturer’s instructions. Samples were analyzed from three independent experiments.

Indirect in-house histo-ELISA was used to measure the production of Col1, α-SMA and Ki67 (ref). Cells were grown as previously described and treated with TGF-β1 or PDGF at 10 ng/mL in association or not with Met 5 mM for 72 h. Afterwards, the supernatant was removed, and wells were washed twice with NaCl and fixed and permeabilized with cold 99% ethanol at room temperature for 10 min. Then, the wells were dried overnight. Cells were incubated overnight with primary antibodies against either human Col1, α-SMA or Ki67. Rabbit or mouse anti-goat peroxidase (Bio-Rad, Hercules, CA, USA) was used as a secondary antibody. Wells were washed with PBS two times, and the horseradish peroxidase activity was revealed using 3,3′,5,5′-tetramethylbenzidine (TMB) solution (Invitrogen, Carlsbad, CA, USA) as HRP substrate and stopped with HCl 0.1 M. Absorbance was read at 450 nm, with a reference at 630 nm, using an 800TS microplate reader (Biotek Cytation 5 imaging reader).

### 2.6. qRT-PCR (Sybergreen) Analyses

Total RNA from HPMSCs, exposed to TGF-β1 or PDGF at 10 ng/mL in association or not with Met 5 mM for 72 h, was extracted directly from harvested cell culture (in 6-well plates) using the Zymo Kit (ZYMO, Irvine, CA, USA, Catalog No.: R1035). A 200 µL volume of RNA Shield and 800 μL of lysis buffer were added to each well, collected and kept at −20 °C until use. qRT-PCR experiments were performed using the One Step Bioline Sensifast Probe NO-ROX One Step Kit (Meridian Bioscience, Cincinnati, OH, USA, Bio-76005) containing the SYBR Green reagent (Lonza, Basel, Switzerland, Cat. No.: 50513). qRT-PCR was performed in a final volume of 5 μL containing 1 μL of extracted total RNA per reaction, 2.7 μL of enzyme mix and 1.3 μL of primer mix, with a final primer concentration of 250 nM. qRT-PCR was carried out in QuantStudio 5 PCR thermocycler (Thermo Fisher Scientific). Relative gene expression was calculated using GAPDH as a reference gene. Experiments were performed in triplicate. Primer and probe sequences related to the genes are listed in Table 1 below.

miRNA from HPMSCs, exposed to TGF-β1 or PDGF at 10 ng/mL in association or not with Met 5 mM for 72 h, was extracted directly from harvested cell culture (in 6-well plates) using the miRNeasy Serum/Plasma Advanced Kit (Ref.: 217204). A 250 µL volume of NaCl was added to each well, collected and kept at −80 °C until use. RT experiments were performed using the miScript II RT (Ref.: 218161). RT was performed in a final volume of 20 μL containing 5 μL of extracted total RNA per reaction and 15 μL of enzyme mix. RT was carried out in the QuantStudio 5 PCR thermocycler (Thermo Fisher Scientific). cDNA was collected and kept at −20 °C until use. qPCR experiments were performed using the miScript SYBR Green PCR (Ref.: 218075). qPCR was performed in a final volume of 5 μL containing 1 μL of extracted cDNA per reaction, 3 μL of enzyme mix and 1 μL of primer mix, with a final primer concentration of 1.25 µM. qPCR was carried out in the QuantStudio 5 PCR thermocycler (Thermo Fisher Scientific). Relative gene expression was calculated using Ce39 as a reference gene. Experiments were performed in triplicate. Primer sequences related to the genes are listed in Table 2 below.

### 2.7. Statistical Analysis

Data were expressed as means ± SEM. All assays were performed in triplicate independent experiments. Statistical analysis was achieved using GraphPad Prism 6 software. Significant differences (*p* < 0.05) between the means were determined by analysis of variance (ANOVA) procedures followed by a multiple comparison test (Bonferroni).

## 3. Results

### 3.1. Study of the Cytotoxicity of the Different Treatments to Induce and Control Fibrosis in Primary Human Pulmonary MSC

Cell viability of HPMSCs under the effect of all treatments was evaluated. For this purpose, the cells were treated with TGF-β1 or PDGF at 10 ng/mL in association or not with Met 5 mM for 72 h. Then, the mitochondrial metabolic activity of the cells was measured by an MTT assay, and to determine the viability of the cells, an LDH release assay was performed. As shown in Figure 1A, all treatments did not affect the mitochondrial metabolic activity of the cells. Basal LDH release was found in the culture supernatants of untreated cells (CT) (Figure 1B). All treatments had no significant effect on LDH release.

### 3.2. Study of Metformin Effects on TGF-β1-Mediated MSC Transdifferentiation into Myofibroblasts

During pulmonary fibrosis, proliferation and differentiation of fibroblasts into myofibroblasts can be observed. These express high levels of alpha-smooth muscle actin (encoded by Acta2), Col1, Col2 and Col4 [12]. Quantitative RT-PCR on HPMSCs stimulated with recombinant TGF-β1 (profibrotic) or PDGF (nonfibrotic) in association or not with Met (potential synthetic antifibrotic agent) was performed to assess the expression of differentiation markers. Treatment with TGF-β1 but not PDGF induced an increase in the expression of all differentiation markers in fibroblasts (Figure 2A–D) by 5-fold for Acta2 as compared to the control, 3-fold for Col1, 1.7-fold for Col2 and 3.7-fold for Col4. Met had no significant effect on the expression of these genes when administered alone. TGF-β1 induced differentiation of fibroblasts into myofibroblasts, and Met exerted an antifibrotic effect by reducing it. In contrast, Met had an effect only on Col2 expression when HPMSCs were treated with PDGF.

The same effects were observed when we tested the expression of collagen 1 at the protein level (Figure 3) and collagen 1 (Figure 4A), α-SMA (Figure 4B) and Ki67 (Figure 4C) at the protein levels by immunofluorescence. TGF-β1 induced an increase in Col1 expression (green) in HPMSCs as compared to control cells (CT). Remarkably, Met treatment showed a reduction in Col1 expression. PDGF appears to have weakly reduced Col1A1 expression. In association with Met, this expression seems to be restored to a basal level (Figure 3).

We also studied the expression of pro- or antifibrotic miRNA in HPMSCs in response to TGF-β1. The expression of miRNA 199 (Figure 5A), miRNA 214 (Figure 5B), miRNA 21 (Figure 5C) was increased in response to TGF-β1 and was reduced for miRNA 29 (Figure 5D). Interestingly, Met was able to significantly reverse the profibrotic effect of TGF-β1.

Matrix metalloproteinases (MMPs) during pathological fibrosis will strongly degrade the extracellular matrix (ECM). [45]. In humans, there are 24 MMPs, 8 of which seem to be directly involved in the fibrosis process [46]. The MMPs most involved in pulmonary fibrosis are MMP1, 2, 3 and 9 [47,48,49]. TGF-β1 induced an increase in the expression of MMP1, 2, 3 and 9 (Figure 6) compared with the control, respectively (from 0.18 ± 0.05 to 0.72 ± 0.05, 2.39 ± 0.08 to 4.37 ± 0.37, 0.01 ± 0.00 to 0.07 ± 0.01, 3.99 × 10^−4^ ± 1.53 × 10^−4^ to 1.59 × 10^−3^ ± 5.93 × 10^−5^). MMP1 expression was increased by PDGF treatment (from 0.18 ± 0.05 to 2.10 ± 0.31). Met alone had no effect on the expression of all genes tested. However, Met was able to inhibit the profibrotic action induced by TGF-β1 and PDGF.

### 3.3. Study of MSC Transdifferentiation into Lipofibroblasts or Osteoblasts

We studied the expression of several genes involved in the differentiation of HPMSCs into adipocytes and osteoblasts in order to shed light on the mechanism of transdifferentiation. TGF-β1 and PDGF are mainly involved in the differentiation process. Indeed, they will allow MSCs expressing several canonical and functional markers, such as CD90 and CD248, to detach from endothelial cells, proliferate and differentiate into lipofibroblasts expressing PPAR-γ, PLIN-2 or calcifying osteoblastic cells expressing high levels of RUNX2 and SOST [50]. We demonstrated that TGF-β1 significantly decreased the expression of CD248 and CD90 compared with the control (Appendix A). TGF-β1 and PDGF significantly increased PDGF expression, respectively (from 1.24 × 10^−2^ ± 2.31 × 10^−3^ to 3.27 × 10^−2^ ± 2.37 × 10^−3^ and 2.23 × 10^−2^ ± 9.85 × 10^−4^) (Figure 7A). Interestingly Met was able to reverse the regulated expression of PDGF.

Figure 7 shows that PDGF can differentiate HPMSCs toward a lipogenic profile by inducing a significant increase in PPAR-γ (Figure 7B) and PLIN2 (Figure 7C) expression compared to control, respectively (from 1.29 × 10^−3^ ± 2.24 × 10^−4^ to 3.15 × 10^−3^ ± 7.92 × 10^−4^ and 8.86 × 10^−2^ ± 6.47 × 10^−3^ to 1.39 × 10^−1^ ± 1.58 × 10^−2^). Met treatment is able to reverse this differentiation. PDGF and TGF-β1 will induce differentiation of HPMSCs to an osteoblastic profile by significantly increasing the RUNX2 genes for PDGF and SOST for TGF-β1 and PDGF. Then, Met treatment is able to reverse this differentiation.

### 3.4. Study of Met Effects on Canonical Smad mRNA Expression

According to several studies, regulation of MSC differentiation may occur via the TGF-β1/Smad signaling pathway [51]. Thus, we investigated whether Met could regulate the expression of the different Smad signaling molecules in HPMSC. Figure 8A shows that TGF-β1 and PDGF had no effect on SMAD2 expression. TGF-β1 induced a decrease in SMAD3 expression (Figure 8B) and a significant increase in SMAD7 expression (Figure 8D), TGF-β1 and PDGF induced a significant increase in SMAD4 expression (Figure 8C) as compared to the control. Met exerted an action by counteracting the effects of TGF-β1 and PDGF.

### 3.5. Study of Met Effects on the Expression of the Pro-Angiogenic Factor VEGF

VEGF plays an essential role in angiogenesis and the development of pathological fibrosis [52]. TGF-β1 induced a significant increase in VEGF expression compared with the control (from 1.91 × 10^−1^ ± 2.65 × 10^−2^ to 4.45 × 10^−1^ ± 6.21 × 10^−2^) (Figure 9). Met had antiangiogenic activity by inhibiting the effect of TGF-β1. PDGF had no significant effect on VEGF mRNA expression.

### 3.6. Study of Met Effects on the Expression of the Growth Factor/Chemokine CXCL-12

During the repair of tissue damage and to allow the recruitment of stem cells a chemokine, called CXCL-12 (i.e., SDF1-alpha) is involved. [53,54]. As shown in Figure 10A and B, TGF-β1 and PDGF induced a significant decrease in the expression of CXCL-12 mRNA and protein. Met did not restore the mRNA expression (Figure 10A) and secretion (Figure 10B).

### 3.7. Study of Met on the Physiological Hormonal Function of MSC

Hormones seem to have a major role in the development of pathological fibrosis. Indeed, calcitonin allows the regulation of collagen and erythropoietin (EPO) and control of programmed cell death [10,55]. TGF-β1 significantly increased the expression of EPO (Figure 11A) and calcitonin (Figure 11B) as compared to the control. Interestingly, Met inhibited the action of TGF-β1. No effect was observed for PDGF.

### 3.8. Study of Met on the Control of AMPK and Pi3K

A reversion of the fibrosis process is possible during tissue healing. During this phenomenon, myofibroblasts are eliminated by apoptosis and thus limit the excessive deposition of the ECM [56]. Apoptosis can be inhibited by TGF-β1, which stimulates the phosphoinositide 3-kinase (Pi3K) pathway [57]. AMP-activated protein kinase (AMPK) regulates cellular metabolism by increasing the production of new mitochondria and normalizes the sensitivity of pulmonary MSCs to apoptosis [42]. Our results show that TGF-β1 and PDGF significantly increased the expression of Pi3K (Figure 12A) and AMPK (Figure 12B) as compared to the control (from 7.67 × 10^−2^ ± 1.06 × 10^−2^ to 2.77 × 10^−1^ ± 1.91 × 10^−2^ and 2.38 × 10^−1^ ± 2.41 × 10^−3^, respectively, and from 1.07 × 10^−1^ ± 7.56 × 10^−4^ to 1.46 × 10^−1^ ± 3.69 × 10^−3^ and 1.30 × 10^−1^ ± 4.54 × 10^−3^). Met was able to inhibit the detrimental effect of TGF-β1 and PDGF by decreasing the expression of Pi3K and AMPK.

### 3.9. Study of Oxidative Stress

During pulmonary fibrosis, oxidative stress is exacerbated [58]. In pulmonary tissues, ROS are produced by NADPH enzymes and mainly by Nox4, and the excessive production of ROS will contribute to the development of pulmonary fibrosis by participating in the differentiation of MSCs into pulmonary myofibroblasts [34]. TGF-β1, which is a profibrotic factor, induces overexpression of Nrf2 [59]. This suggests a role for the transcription factor Nrf2 in the modulation of the cellular phenotype during fibrosis processes and its probable interest as a therapeutic target [35]. In our study TGF-β1 induced overexpression of Nox4 (Figure 13A) and decreased expression of Nrf2 (Figure 13B) as compared to the control. Met inhibited the action of TGF-β1 on Nox4 expression but had no effect on Nrf2 expression.

## 4. Discussion

Pathological pulmonary fibrosis is a chronic process in which signaling molecules and the TGF-β pathway are involved. It is characterized by an excessive proliferation of fibroblasts, a differentiation and a deposition of the ECM linked to the accumulation of myofibroblasts [36,60,61]. Pulmonary fibrosis also develops due to the mobilization of MSCs and their differentiation into myofibroblasts [1,62,63]. Despite the progress made in the treatment of fibrosis, there is still no effective and safe therapeutic approach [64]. Thus, the discovery and exploration of new treatments against pathological fibrosis are of major importance.

In our study, we explored the therapeutic potential of metformin (a synthetic biguanide derived from galegine extracted from the Fabaceae *Galega officinalis*) in the attenuation of pulmonary fibrosis by inhibiting proliferation, differentiation and ECM production of myofibroblasts. Metformin is known to have antidiabetic properties but has never been tested in the context of an in vitro model of MSC-mediated fibrosis.

The novel antifibrotic activities of metformin, as well as many natural products such as quercetin, resveratrol or berberine, have been linked to their capacity to negatively interfere with the mitochondrial complex 1 molecule and by limiting the rate of ATP production [65,66,67]. Among the signaling pathways involved, this inhibition will lead to the activation of AMPK, a major sensor of cellular energy. Metformin, as well as direct activators of AMPK (e.g., AICAR), has been shown to inhibit collagen production in primary human lung fibroblasts and, interestingly to enhance transdifferentiation of aggressive myofibroblasts to lipofibroblasts related to phenotypic recovery from fibrosis [42].

First and foremost, we validated our in vitro model of lung fibrosis following the treatments of primary HPMSCs with recombinant TGF-β1. Indeed, our MSC fibroblast-like cells expressing the canonical markers CD248 and CD90 were able to differentiate into myofibroblast key effector cells of pulmonary fibrosis [68]. Myofibroblasts expressed high levels of Acta2 (coding for alpha SMA) and collagens such as Col1 A1, Col 2 Col 4 (Figure 2), as described in the literature [69]. These are major components of the ECM [70]. While Met alone had no effects on the expression of the different ECM genes, they all had notable inhibitory effects on the TGF-β1-stimulated expression of Acta2 and Col AI, Col II and Col IV mRNA. This observation was confirmed at the protein level for collagen (Figure 3). MSCs can also respond to other growth factors such as PDGF and, for instance, in an autocrine manner (as validated in Figure 7A), but we found no effects of Met on PDGF-mediated activities.

Among the key mechanisms controlling MSC differentiation into myofibroblasts, miRNAs are also key players. Studies have shown that miRNA could be involved in fibrogenesis, especially miRNA 21 [23,71], miRNA 199 and 214 [72]. In contrast, the expression of miR-29 is significantly reduced in fibrotic lungs [73]. Our results (Figure 5) show that TGF-β1 upregulated the expression of all three profibrotic miR and downregulated miR29 expression. Interestingly, Met was able to drastically control TGF-β1-regulated expression of miR 21,199 and 214 but upregulated the expression of miR29 and in line with previous findings [74]. Several miRs have been associated with the control of fibrosis, and a more comprehensive analysis of Met effects on the miRNome of MSCs is now highly warranted. Of note, we have tested four miRs as prototypical markers of myofibroblasts, and our results confirmed that Met can control the process of fibrosis at least in vitro.

Among the members of the matrix metalloproteinase, MMP-9 is particularly involved in the development of pulmonary fibrosis [75,76]. In our study, we found that the expression of MMP-1, 2, 3 and 9 was increased by TGF-β1 stimulation (Figure 5). Met inhibited TGF-β1-induced overexpression of the various metalloproteinases. The expression of MMP-1 was particularly increased by PDGF. MMP-1 breaks down collagens 1 and 2, and we found that Met was able to downregulate the overexpression of MMP-1 induced by PDGF. PDGF has proliferative effects on MSCs, but it is also involved in fibrosis by upregulating the expression of TGF-β1 by MSCs and, in return, stimulating PDGF expression, as verified in Figure 7A. This is a major autocrine and profibrotic loop that can be controlled by Met (Figure 7A).

Metformin exhibited an ability to transdifferentiate myofibroblasts into lipofibroblasts [42]. Several studies have shown that TGF-β1 in the context of pulmonary fibrosis downregulated PPAR-γ signaling [77,78]. In our HPMSC model, we found that TGF-β1 had a modest effect on the expression of PPAR-γ and PLIN2 (Figure 7B,C). Conversely, PDGF alone induced upregulated expression of PPAR-γ and PLIN2, promoting a lipofibroblast profile. Hence, PDGF may represent a molecular regulator to switch between an aggressive myofibroblast profile to a more homeostatic lipofibroblast profile. Metformin did not regulate the expression of PPAR-γ and PLIN2 but reversed the effect of PDGF. This unique observation goes against the therapeutic potential of Met in fibrosis but should be considered in the context of another chronic pathology where the transdifferentiation of myofibroblasts towards adipocyte-like cells is critically important. Indeed, it would be interesting to evaluate the potential beneficial effects of Met in atherosclerosis associated with excessive fat depots in the arteries.

MSCs are common progenitor cells of adipocytes, as well as osteoblasts, and they are delicately balanced for their differentiation commitment. A variety of external cues contribute to the delicate balance of adipo-osteogenic differentiation of MSCs, including chemical, physical and biological factors. Our study also investigated the role Met may exert on TGF-β and PDGF potential to control the differentiation of HPMSCs into osteoblast-like cells by addressing the levels of expression of canonical markers RUNX2 and SOST/sclerostin (Figure 7). The positive regulation was particularly marked for SOST under TGF-β1, while Met downregulated this effect. We will need to explore other markers of osteoblasts, such as osterix/SP7 and RANKL. This is an important and warranted investigation to address the therapeutic potential of Met in the ill-characterized pathological process of vascular calcification [79].

TGF-β1 initiates canonical and noncanonical pathways to engage multiple biological effects. Among them, SMAD signaling is recognized as a major pathway. During fibrogenesis, SMAD3 is highly activated, which is associated with the downregulation of the inhibitory activity of SMAD7. The equilibrium shift between SMAD3 and SMAD7 leads to accumulation and activation of myofibroblasts and overproduction of ECM [80]. Overexpression of SMAD7 is known to be of therapeutic agent potential. Focusing only on the level of expression and not activation, we found that TGF-β1 alone upregulated the expression of SMAD7 and downregulated that of SMAD3. It is important to note that the level of SMAD3 is at least tenfold higher compared to SMAD7 and hence promoting myofibroblast differentiation. Counterintuitively, our results (Figure 8) showed that Met attenuated the effects of TGF-β1 on SMAD7.

The endothelium participates In the maintenance of pulmonary fibrosis, notably by facilitating angiogenesis [81]. One of the key players in angiogenesis and well known to participate in fibrosis is VEGF. In mouse models, a high severity of fibrosis is associated with an increase in angiogenesis via an accumulation of VEGF in the lungs [82]. Our results demonstrated that TGF-β1 increased VEGF expression (Figure 9). This effect was equally reversed by the addition of Met to TGF-β1 treatment. VEGF is produced by perivascular MSCs to ensure endothelium homeostasis. This is also the case for the other growth factor CXCL12 (SDF1-α), but which is less involved in the process of fibrosis. In the bone marrow, CXCL12 produced by MSCs contributes to the retention of hematopoietic stem cells [83]. We found no significant effects of Met on its expression at the mRNA and protein levels (Figure 10). Perhaps less emphasized is the capacity of MSCs to maintain tissue homeostasis through the expression of key hormones such as calcitonin and EPO [10]. Interestingly, the expression of both was upregulated by TGF-β1 treatment of HPMSCs, and the effect was natively controlled by Met. This observation further illustrates that several actors of fibrosis downstream of the TGF-β signaling receptor are equally affected by Met.

The PI3 kinase pathway contributes to TGF-β1 induced fibrosis via Akt and PAK2 and has a key role in activation of MSCs and myofibroblast survival [84]. Our results show that TGF-β1 increases Pi3K expression (Figure 12A). Met has an antifibrotic effect by inhibiting the effect of TGF-β1. Conversely, it has been shown that AMPK inhibits the TGF-β1-induced fibrogenic property of MSCs of the liver (i.e., hepatic stellate cells) by regulating transcriptional coactivator p300 [85]. We found that TGF-β1 significantly upregulated the expression of AMPK mRNA (Figure 12B). It remains to be tested whether this transcriptome regulation can be extended to the level of AMPK phosphorylation.

Finally, our study points to a major role of Met in controlling Nox expression implicated in the oxidative stress response of TGF-β1-activated myofibroblasts. Waghray et al. reported in 2005 that myofibroblast produced high levels of hydrogen peroxide (H_2_O_2_), a major diffusible signal for the induction of fibrosis, as well as cell death and tissue injuries [86]. In 2009, Hecker et al. attributed this observation to the critical role of TGF-β1 to upregulate (by many folds) the expression of Nox4 by pulmonary MSCs [87]. The authors have shown that Nox4 knockdown drastically reduces (H_2_O_2_) myofibroblast differentiation, extracellular matrix (ECM) production and contractility of tissues. Our HPMSC model confirmed the major effects of TGF-β1 in the regulation of Nox4 and which were largely impacted when cells were cotreated with Met (Figure 13). Remarkably, this is in line with previous observations [88]. Indeed, Sato and colleagues reported in 2016 that TGF-β1-induced myofibroblast differentiation was clearly inhibited by metformin treatment. Metformin-mediated indirect activation of AMPK (implicating the inhibition of the mitochondrial complex I) was involved in the inhibition of TGF-β1-induced Nox4 expression. Nox4 knockdown and N-acetylcysteine treatment illustrated that Nox4-derived ROS generation was critical for TGF-β-induced SMAD phosphorylation and myofibroblast differentiation [88]. Nrf2 transcription factor is a key regulator in the cellular response to oxidative stress by controlling the expression of antioxidant molecules. A decrease in Nrf2 associated with an upregulation of Nox4 leads to an increase in oxidative stress in fibrotic lungs [87]. Pulmonary fibrosis, therefore, involves oxidative stress, persistence of myofibroblasts and resistance to apoptosis as a result of Nox4/Nrf2 imbalance [89]. In our study (Figure 13B), we found that TGF-β1, while upregulating NOX4, had the opposite effect on Nrf2 expression, as previously described [89]. Interestingly, Met restored Nrf2 expression.

## 5. Conclusions

In summary (outlined in Figure 14), our data demonstrated for the first time the metformin effect against TGF-β1-induced fibrosis in HPMSC. Met attenuated the fibrotic process by interfering with TGF-β1/SMAD activities, notably through the regulated expression of ECM and by controlling the Nox4/Nrf2-mediated redox imbalance. The effects of Met promoting transdifferentiation (a process also called fibrosis reversion) of myofibroblasts towards lipofibroblasts and osteoblasts is promising, and further experiments along these lines are now highly warranted in vitro but also benefiting from an animal model of lung fibrosis. We are convinced that Met should be considered as a possible treatment for patients with pathological pulmonary fibrosis in addition to its application for diabetics.

## Figures and Tables

**Figure 1 cells-11-04090-f001:**
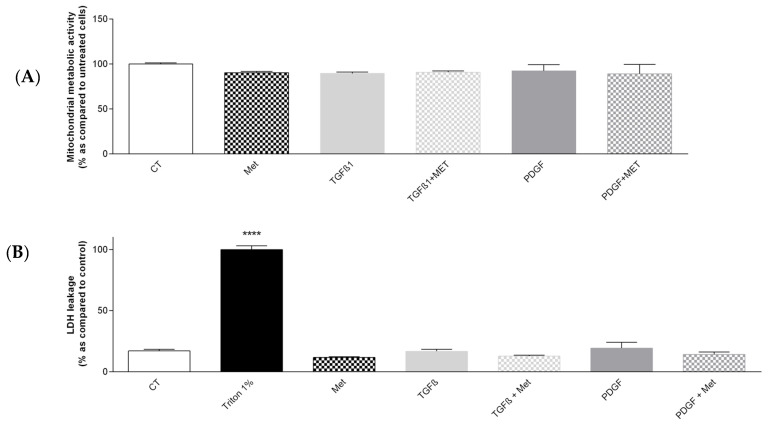
Results of the mitochondrial metabolic activity and LDH release from HPMSCs exposed to TGF-β1 or PDGF in association or not with metformin (Met). Cells were exposed to TGF-β1 or PDGF at 10 ng/mL in association or not with Met (5 mM) for 72 h. (**A**) Mitochondrial metabolic activity was determined by MTT assay. (**B**) Released LDH in culture medium was measured using a colorimetric-based method (CytoTox 96^®^ Non-Radioactive Cytotoxicity Assay) and expressed relative to the maximum release by application of a lysis buffer (Triton 1%). Reported values are means ± SEM of three independent experiments, and *p*-value was calculated using the Bonferroni multiple comparison test (****: *p* < 0.0001) as compared to control.

**Figure 2 cells-11-04090-f002:**
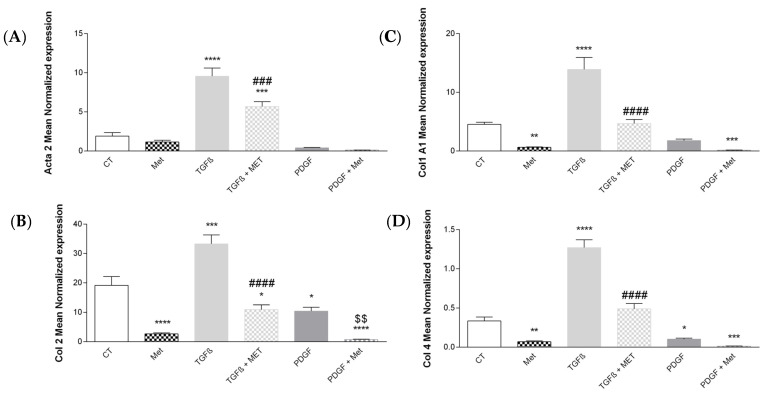
Differentiation of HPMSCs into myofibroblast-like cells (higher expression of Acta2 and collagens) is inhibited by metformin. Cells were exposed to TGF-β1 or PDGF at 10 ng/mL in association or not with Met (5 mM) for 72 h. Then, RNA was collected, and (**A**) Acta2, (**B**) Col2, (**C**) Col1A1 and (**D**) Col4 gene expression levels were determined by qRT-PCR. Reported values are means ± SEM of three independent experiments, and *p*-value was calculated using the Bonferroni multiple comparison test: *: *p* < 0.05, **: *p* < 0.01, ***: *p* < 0.001 and ****: *p* < 0.0001 as compared to control, ###: *p* < 0.001 and ####: *p* < 0.0001 as compared to TGF-β1 and $$: *p* < 0.01 as compared to PDGF.

**Figure 3 cells-11-04090-f003:**
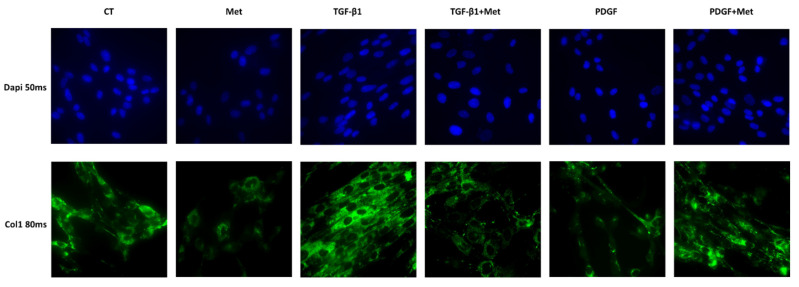
Met reduced the expression of the canonical ECM component collagen protein in TGF-β1-induced myofibroblasts. HPMSCs were exposed to TGF-β1 or PDGF at 10 ng/mL in association or not with Met 5 mM for 72 h. Cells were probed with primary antibody Col1 before nuclei staining by DAPI (blue) and antibody binding by Alexa Fluor^®^ 488-conjugated rabbit anti-mouse IgG antibody (green). Three independent experiments were performed.

**Figure 4 cells-11-04090-f004:**
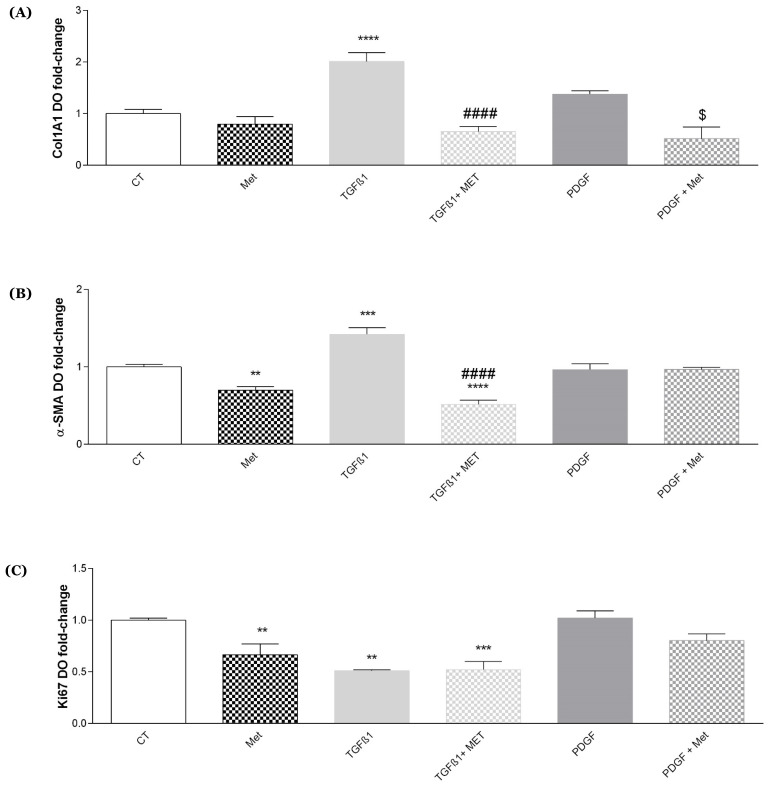
Met altered the expression of collagen, α-SMA and Ki67 protein in TGF-β1-induced myofibroblasts. Cells were exposed to TGF-β1 or PDGF at 10 ng/mL in association or not with Met (5 mM) for 72 h. Then, proteins were collected, and (**A**) Col1A1, (**B**) α-SMA and (**C**) Ki67 levels were determined by indirect in-house ELISA. Reported values are means ± SEM of three independent experiments, and *p*-value was calculated using the Bonferroni multiple comparison test: **: *p* < 0.01; ***: *p* < 0.001 and ****: *p* < 0.0001 as compared to control, ####: *p* < 0.0001 as compared to TGF-β1 and $: *p* < 0.05 as compared to PDGF.

**Figure 5 cells-11-04090-f005:**
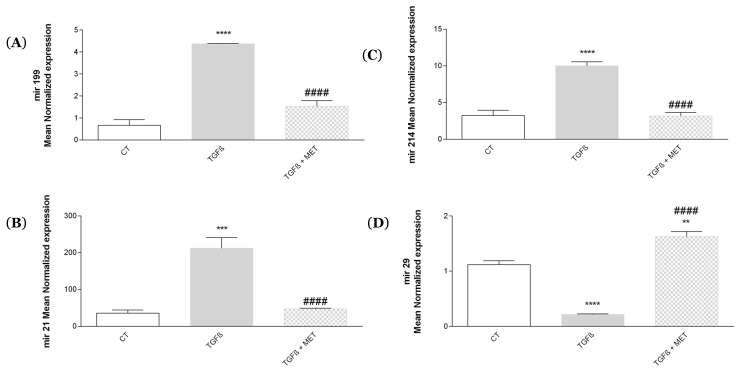
Met regulates the expression of different pro- and antifibrotic miRNA. Cells were exposed to TGF-β1 at 10 ng/mL in association or not with Met (5 mM) for 72 h. Then, cDNA was used and (**A**) miRNA 199, (**B**) miRNA 214, (**C**) miRNA 21 and (**D**) miRNA 29 gene expression levels were determined by qPCR. Reported values are means ± SEM of three independent experiments, and *p*-value was calculated using the Bonferroni multiple comparison test: **: *p* < 0.01, ***: *p* < 0.001 and ****: *p* < 0.0001 as compared to control and ####: *p* < 0.0001 as compared to TGF-β1.

**Figure 6 cells-11-04090-f006:**
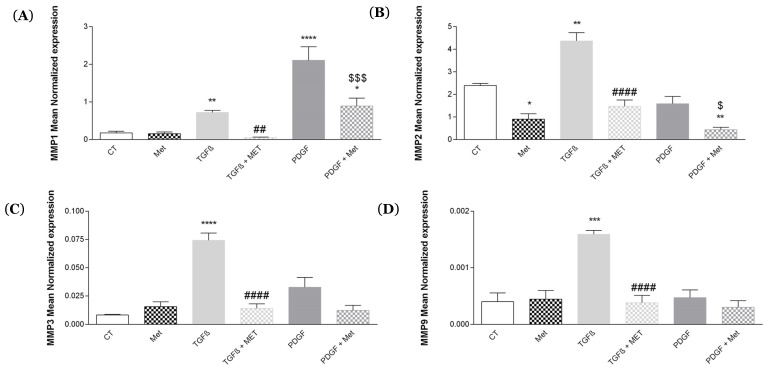
Regulated expression of TGF-β1-induced MMP by Met. Cells were exposed to TGF-β1 or PDGF at 10 ng/mL in association or not with Met (5 mM) for 72 h. Then, RNA was collected, and (**A**) MMP1, (**B**) MMP2, (**C**) MMP3 and (**D**) MMP9 gene expression levels were determined by qRT-PCR. Reported values are means ± SEM of three independent experiments, and *p*-value was calculated using the Bonferroni multiple comparison test: *: *p* < 0.05, **: *p* < 0.01, ***: *p* < 0.001 and ****: *p* < 0.0001 as compared to control, ##: *p* < 0.01 and ####: *p* < 0.0001 as compared to TGF-β1 and $: *p* < 0.05 and $$$: *p* < 0.001 as compared to PDGF.

**Figure 7 cells-11-04090-f007:**
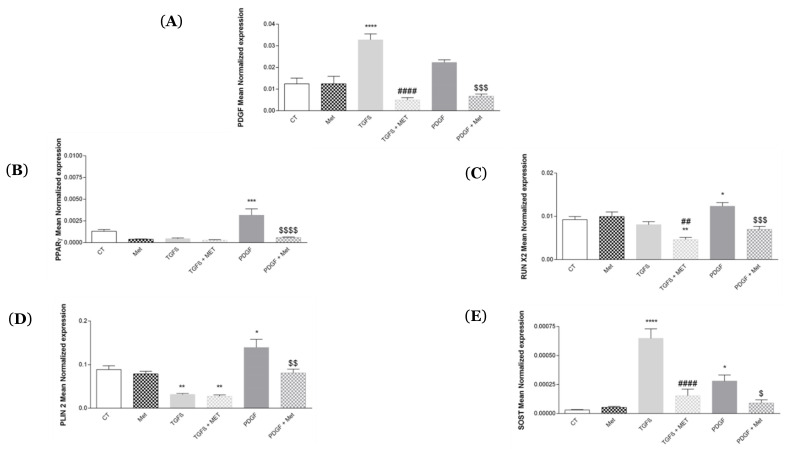
Met regulates MSC transdifferentiation into either lipofibroblasts or osteoblasts. Cells were exposed to TGF-β1 or PDGF at 10 ng/mL in association or not with Met (5 mM) for 72 h. Then, RNA was collected, and (**A**) PDGF, (**B**) PPAR-γ, (**C**) PLIN2, (**D**) RUN X2 and (**E**) SOST gene expression levels were determined by qRT-PCR. Reported values are means ± SEM of three independent experiments, and *p*-value was calculated using the Bonferroni multiple comparison test: *: *p* < 0.05, **: *p* < 0.01, ***: *p* < 0.001 and ****: *p* < 0.0001 as compared to control, ##: *p* < 0.01 and ####: *p* < 0.0001 as compared to TGF-β1 and $: *p* < 0.05, $$: *p* < 0.01, $$$: *p* < 0.001 and $$$$: *p* < 0.0001 as compared to PDGF.

**Figure 8 cells-11-04090-f008:**
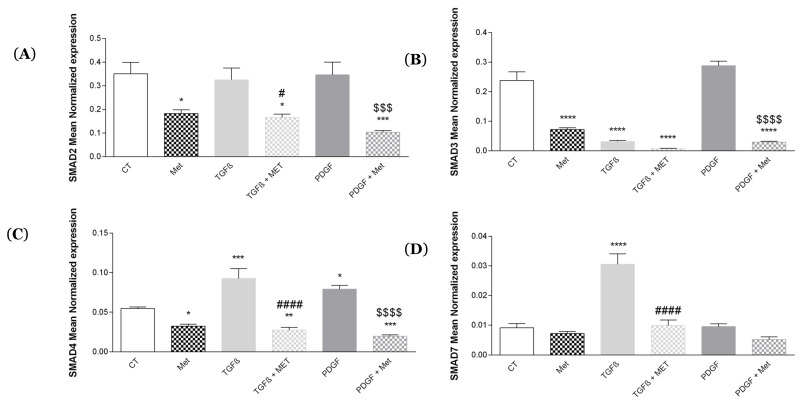
Met regulates the expression of canonical Smad molecules downstream of the TGF-β1 pathway. Cells were exposed to TGF-β1 or PDGF at 10 ng/mL in association or not with Met 5 mM for 72 h. Then, RNA was collected, and (**A**) SMAD2, (**B**) SMAD3, (**C**) SMAD4 and (**D**) SMAD7 gene expression levels were determined by qRT-PCR. Reported values are means ± SEM of three independent experiments, and *p*-value was calculated using the Bonferroni multiple comparison test: *: *p* < 0.05, **: *p* < 0.01, ***: *p* < 0.001 and ****: *p* < 0.0001 as compared to control, #: *p* < 0.05 and ####: *p* < 0.0001 as compared to TGF-β1, $$$: *p* < 0.001 and $$$$: *p* < 0.0001 as compared to PDGF.

**Figure 9 cells-11-04090-f009:**
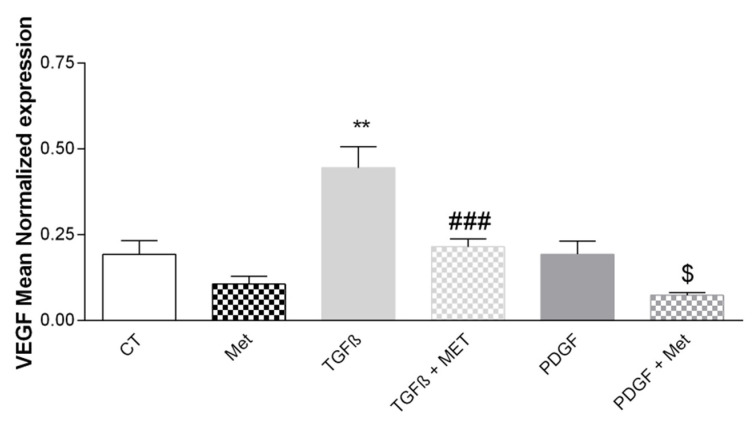
Met regulates the expression of the VEGF proangiogenic factor. Cells were exposed to TGF-β1 or PDGF at 10 ng/mL in association or not with Met 5 mM for 72 h. Then, RNA was collected and VEGF gene expression levels were determined by qRT-PCR. Reported values are means ± SEM of three independent experiments, and *p*-value was calculated using the Bonferroni multiple comparison test: **: *p* < 0.01 as compared to control, ###: *p* < 0.001 as compared to TGF-β1 and $: *p* < 0.05 as compared to PDGF.

**Figure 10 cells-11-04090-f010:**
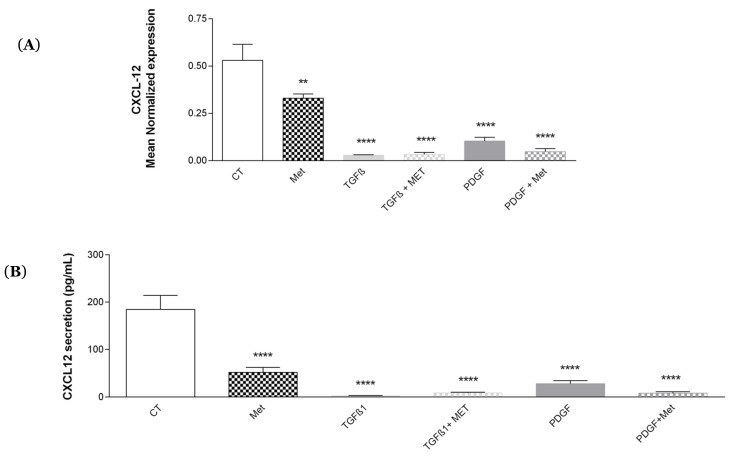
Effects of Met on CXCL-12 (SDF1-alpha) expression and secretion. Cells were exposed to TGF-β1 or PDGF at 10 ng/mL in association or not with Met (5 mM) for 72 h. Then, RNA was collected and CXCL-12 gene expression levels were determined by qRT-PCR, (**A**) RNA level. Then, culture media were collected, CXCL-12 level was evaluated by ELISA, and (**B**) protein levels were tested by ELISA. Reported values are means ± SEM of three independent experiments, and *p*-value was calculated using the Bonferroni multiple comparison test: **: *p* < 0.01 and ****: *p* < 0.0001 as compared to control.

**Figure 11 cells-11-04090-f011:**
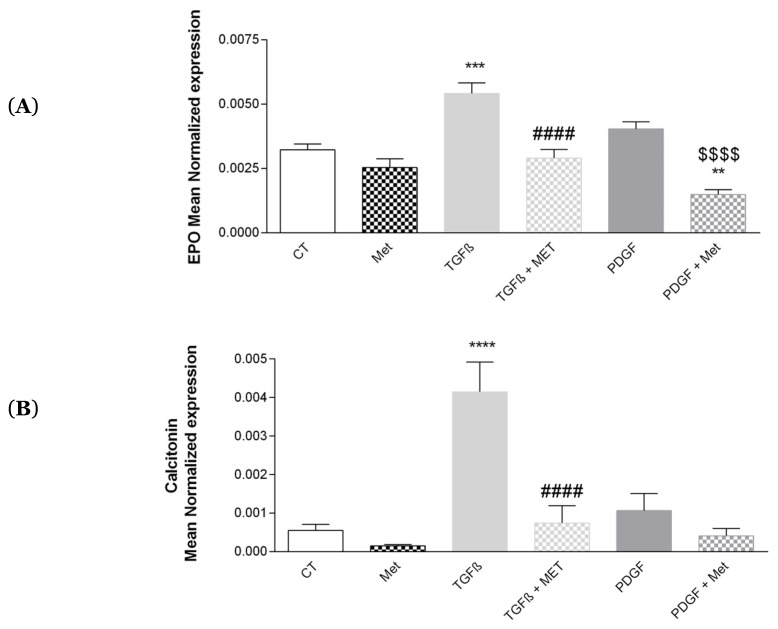
Effects of Met on MSC hormone expression: erythropoietin (EPO) and calcitonin. Cells were exposed to TGF-β1 or PDGF at 10 ng/mL in association or not with Met (5 mM) for 72 h. Then, RNA was collected, and (**A**) EPO and (**B**) calcitonin gene expression levels were determined by qRT-PCR. Reported values are means ± SEM of three independent experiments, and *p*-value was calculated using the Bonferroni multiple comparison test: **: *p* < 0.01, ***: *p* < 0.001 and ****: *p* < 0.0001 as compared to control, ####: *p* < 0.0001 as compared to TGF-β1and $$$$: *p* < 0.0001 as compared to PDGF.

**Figure 12 cells-11-04090-f012:**
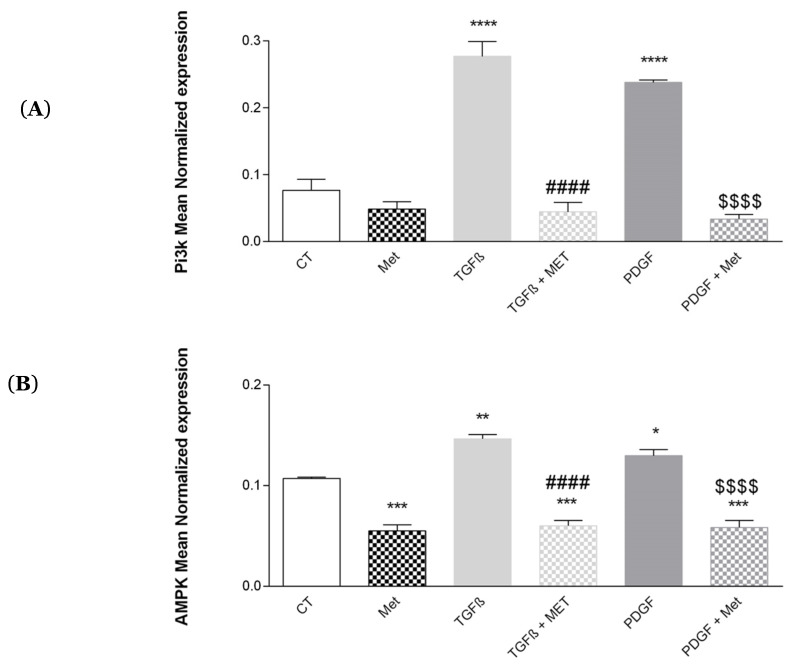
Effects of Met on PI3K and AMPK. Cells were exposed to TGF-β1 or PDGF at 10 ng/mL in association or not with Met (5 mM) for 72 h. Then, RNA was collected, and (**A**) Pi3K and (**B**) AMPK gene expression levels were determined by qRT-PCR. Reported values are means ± SEM of three independent experiments, and *p*-value was calculated using the Bonferroni multiple comparison test: *: *p* < 0.05, **: *p* < 0.01, ***: *p* < 0.001 and ****: *p* < 0.0001 as compared to control, ####: *p* < 0.0001 as compared to TGF-β1 and $$$$: *p* < 0.0001 as compared to PDGF.

**Figure 13 cells-11-04090-f013:**
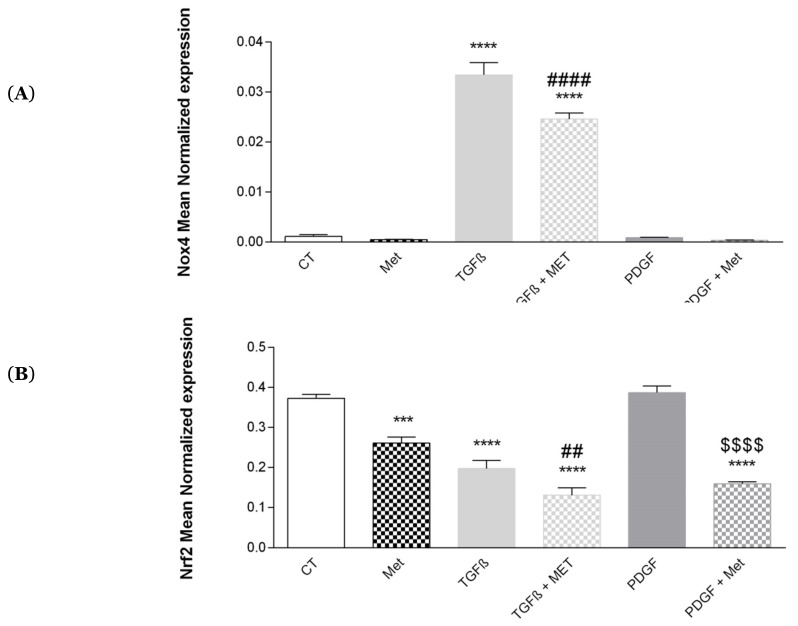
Effects of Met on NRF2 and NOX4 of the oxidative stress response. Cells were exposed to TGF-β1 or PDGF at 10 ng/mL in association or not with Met (5 mM for 72 h). Then, RNA was collected, and (**A**) Nox4 and (**B**) Nrf2 gene expression levels were determined by qRT-PCR. Reported values are means ± SEM of three independent experiments, and *p*-value was calculated using the Bonferroni multiple comparison test: ***: *p* < 0.001 and ****: *p* < 0.0001 as compared to control, ##: *p* < 0.01 and ####: *p* < 0.0001 as compared to TGF-β1 and $$$$: *p* < 0.0001 as compared to PDGF.

**Figure 14 cells-11-04090-f014:**
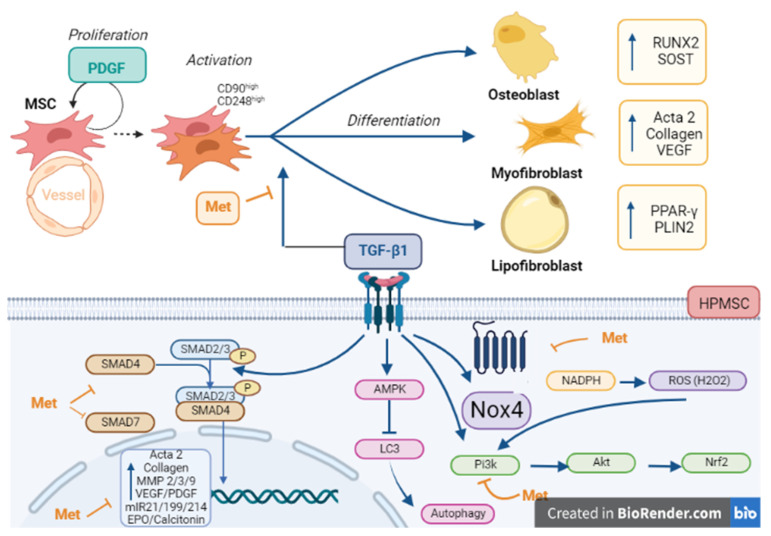
Summary figure on the emerging antifibrotic role of BS in human pulmonary mesenchymal stem cells.

**Table 1 cells-11-04090-t001:** Primers used for qRT-QPCR analysis.

Target Gene	Forward	Reverse	Detection
GAPDH	TGCGTCGCCAGCCGAG	AGTTAAAAGCAGCCCTGGTG	SYBR Green
Acta2	TTCATCGGGATGGAGTCTGCTGG	TCGGTCGGCAATGCCAGGGT	SYBR Green
Col1 A1	TCTAGACATGTTCAGCTTTGTGGAC	TCTGTACGCAGGTGATTGGT	SYBR Green
Col 2	GAGGGCAACAGCAGGTTCACTTA	TCAGCACCACCGATGTCCA	SYBR Green
Col 4	GGACGGCGTAGGCTTCTTG	GCAAACGCTTCAGCTTTTGG	SYBR Green
PPAR-γ	GGTGACCAGAAGCCTGCATT	TGTCAACCATGGTCATTCGTT	SYBR Green
PLIN 2	GCTGCAGTCCGTCGATTTCT	TCACACCGTTCTCTGCCATC	SYBR Green
RUNX 2	TAGGCGCATTTCAGGTGCTT	TGCATTCGTGGGTTGGAGAA	SYBR Green
PDGF	TCCTGTCTCTCTGCTGCTAC	ATCAAAGGAGCGGATCGAGT	SYBR Green
SOST	TTCCCCGGATGTTTGGCTAC	AGTTGGGGCGGATGTGATTT	SYBR Green
SMAD2	AACAGAACTTCCGCCTCTGG	GGAGGTGGCGTTTCTGGAAT	SYBR Green
SMAD3	TGGACGCAGGTTCTCCAAAC	CCGGCTCGCAGTAGGTAAC	SYBR Green
SMAD4	GGACTGTTGCAGATAGCATC	GCTGGAATGCAAGCTCATTG	SYBR Green
SMAD7	CCATCGGGTATCTGGAGTAAGGA	TGCTGTGCAAAGTGTTCAGGTG	SYBR Green
VEGF	ACAACAAATGTGAATGCAGACCA	GAGGCTCCAGGGCATTAGAC	SYBR Green
CXCL-12	CTACAGATGCCCATGCCGAT	CAGCCGGGCTACAATCTGAA	SYBR Green
EPO	CGAGAATATCACGACGGGCT	CAGACTTCTACGGCCTGCTG	SYBR Green
Calcitonin	ATCAGAGACACTGCCCAGC	CCAGGGCAGACCTGAATGG	SYBR Green
MMP1	TTTGTCAGGGGAGATCATCGG	TCCAAGAGAATGGCCGAGTT	SYBR Green
MMP 2	CCCTGATGTCCAGCGAGTG	ACGACGGCATCCAGGTTATC	SYBR Green
MMP 3	TCAGTCCCTCTATGGACCTCCC	GGTTCAAGCTTCCGAGGGAT	SYBR Green
MMP 9	TGCCCGGACCAAGGATACAGTTT	GTTCAGGGCGAGGACCATAGAGG	SYBR Green
Pi3k	TCTTTGTGCAACCTACGTGA	AGCCATTCATTCCACCTGGG	SYBR Green
AMPK	TGTCACAGGCATATGGTGGTC	GGGCCTGCATACAATCTTCC	SYBR Green
Nrf2	GCTATGGAGACACACTACTTGG	CCAGGACTTCAGGCAATTCT	SYBR Green
Nox4	TCGCCAACGAAGGGGTTAAA	GACACAATCTAGCCCCAACA	SYBR Green
CD 90	TGAAAACTGCGGGGTCCGA	TGCAAGACTGTAGCAGGGAG	SYBR Green
CD 248	TTGCACTGGGCATCGTGTA	TTGCTCCCAGCATGGATGAC	SYBR Green

**Table 2 cells-11-04090-t002:** Primers used for miR analysis.

Target Gene	Forward	Detection
hsa miR-21-5p	GCAGTAGCTTATCAGACTGATG	SYBR Green
hsa miR-214-3p	ACAGCAGGCACAGACA	SYBR Green
hsa miR-29b-3p	CAGTAGCACCATTTGAAATCAG	SYBR Green
hsa miR-199a-5p	GCCCAGTGTTCAGACTAC	SYBR Green

## Data Availability

Not applicable.

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
