# Peer review of "Deciphering the Antifibrotic Property of Metformin"

_cells, 2022, doi:10.3390/cells11244090_

Round 1

Reviewer 1 Report

The article # cells-1990824 by Axelle et al. is trying to explore the antifibrotic property of metformin (Met).  The authors, through results from different in-vitro experiments, are trying to reinforce or reiterate the concept which is not new to the field. However, the overall volume of experiments or data collected meets the target to draft a decent manuscript for publication purposes. The results graph is clear with all proper controls included. However, the article has a few problems with writing errors. In a few instances, fluency in English makes the presentation of the text unclear, I recommend a thorough check of the writing of the whole text.

This reviewer has the following comments.

1.     Have the author tried some lower doses of metformin (Met) in any of their experimental setups? As a scientist, we look for drugs with a lower treatment dose that gives a high efficacy or potency.

2.     The results and the corresponding conclusions were primarily drawn from in-vitro experiments. If the author can manage to report similar findings from in-vivo experiments, the quality or the scientific merits of this manuscript could increase many-fold. Authors should think of a few supporting ex-vivo experiments.  It is a must. 

3.     The title of section 3.3 is in French. Kindly revert to English .

Author Response

Dear Reviewer,

Thanks for your thorough review of this manuscript.

We really appreciate the helpful comments from the reviewers. The questions raised by reviewers are answered just following the comments.

Reviewer 1:

The article # cells-1990824 by Axelle et al. is trying to explore the antifibrotic property of metformin (Met).  The authors, through results from different in-vitro experiments, are trying to reinforce or reiterate the concept which is not new to the field. However, the overall volume of experiments or data collected meets the target to draft a decent manuscript for publication purposes. The results graph is clear with all proper controls included. However, the article has a few problems with writing errors. In a few instances, fluency in English makes the presentation of the text unclear, I recommend a thorough check of the writing of the whole text.

We would like to thank reviewer 1 for his comments and corrections on our manuscript.

  1. Have the author tried some lower doses of metformin (Met) in any of their experimental setups? As a scientist, we look for drugs with a lower treatment dose that gives a high efficacy or potency.

Thank you for this comment.

We performed a 72h LDH test to determine the cytotoxicity of metformin. 5mM was the highest dose with no effect on HPMSC. Also, for the use of the 5mM concentration we based ourselves on the literature. At this dose the effects are more marked after 72h of treatment (1).

(1).  Kheirollahi V, Wasnick RM, Biasin V, Vazquez-Armendariz AI, Chu X, Moiseenko A, et al. Metformin induces lipogenic differentiation in myofibroblasts to reverse lung fibrosis. Nat Commun. 5 juill 2019;10(1):2987.

  1. The results and the corresponding conclusions were primarily drawn from in-vitro experiments. If the author can manage to report similar findings from in-vivo experiments, the quality or the scientific merits of this manuscript could increase many-fold. Authors should think of a few supporting ex-vivo experiments. It is a must.

Thank you for this comment.

In the present study, we investigated whether metformin could have antifibrotic effects on human lung mesenchymal stem cells with a TGFb1-induced profibrotic differentiation phenotype. This study is not well described in the literature. And this is where our study is new. This is a preliminary study that will serve as a model for future in vivo studies. We want to use metformin as a positive control and to do so we had to go through in vitro analyses. To date we have no in vivo data. However, the literature is provided on this subject for the treatment of pulmonary fibrosis but not on the fate of stem cells (2).

(2). Gu X, Han YY, Yang CY, Ji HM, Lan YJ, Bi YQ, et al. Activated AMPK by metformin protects against fibroblast proliferation during pulmonary fibrosis by suppressing FOXM1. Pharmacological Research. 1 nov 2021;173:105844.

  1. The title of section 3.3 is in French. Kindly revert to English.

Study of MSC transdifferentiation into lipofibroblasts or osteoblasts. Modifications have been made.

The manuscript has been checked and corrected. Thanks for thorough review of this manuscript.

Reviewer 2 Report

In this paper, the authors describe a comprehensive work on anti-fibrotic effects of metformin using TGFb1/PDGF-induced fibrosis in human pulmonary mesenchymal stem cells (HPMSC).  However, the findings are descriptive in nature examining up- and down-regulation of a number of pro-fibrotic, MSC differentiation, oxidative and anti-oxidative stress markers.

Major Comments:

1. Introduction, as written, is unfocused and largely diffuse with a broad literature review of fibrosis research.  The main focus of the paper is detailed at the end, but it suffers a clear rationale for the study.  The authors needs to provide more scientific rationale to study the effects of metformin in pulmonary fibrosis and, in particular, in HPMSC.

2. The main focus seems to be on TGFb1-induced fibrosis model, but it is not all clear on the use of PDGF.

3. Met was used at 5mM for 72h.  It would be nice to have a concentration and time response effects.

4. It appears multiple comparisons were tested against controls only, followed by post hoc analysis of different treatment groups (i.e., TGFb vs TGFb+Met).  Are the P values adjusted for multiple comparisons?

5. Figure 3 shows the effects of PDGF and PDGF+Met, but the results are not described in the main text. Also, the corresponding bar graphs are missing.  It appears that Met decreased Col1 expression, as did for PDGF. But, PDGF+Met seems to increase Col1.  Please comment on this.

6. Need to provide more details on the pro- or anti-fibrotic miRNAs (miR199, miR214, and miR21) and their specific or reported target mRNAs for post-transcriptional gene regulation. Again, what is the effect of PDGF on these miRs?  Would anti-miRs sufficient to inhibit the action of TGFb?

7. Figure 8. It appears PDGF is equally strong activator of SMAD signaling as TGFb, yet the effects of PDGF on all indices measured related to fibrosis are mild/low.  Please comment.

8. Met is an activator of AMPK. Figure 12B is antithetic to its known role.

9. Most of the markers were examined using RT-PCR.  How about the protein levels.  On that note, does Met have an effect on Nrf2 activity?

10. HPMSC was used throughout. How about fibroblasts or myofibroblasts?

Author Response

Dear Reviewer,

Thanks for your thorough review of this manuscript.

We really appreciate the helpful comments from the reviewers. The questions raised by reviewers are answered just following the comments.

Reviewer 2:

In this paper, the authors describe a comprehensive work on anti-fibrotic effects of metformin using TGFb1/PDGF-induced fibrosis in human pulmonary mesenchymal stem cells (HPMSC).  However, the findings are descriptive in nature examining up- and down-regulation of a number of pro-fibrotic, MSC differentiation, oxidative and anti-oxidative stress markers.

We would like to thank reviewer 2 for his comments and corrections on our manuscript.

  1. Introduction, as written, is unfocused and largely diffuse with a broad literature review of fibrosis research. The main focus of the paper is detailed at the end, but it suffers a clear rationale for the study. The authors need to provide more scientific rationale to study the effects of metformin in pulmonary fibrosis and, in particular, in HPMSC.

Thank you for this very accurate comment. We have added a paragraph in the introduction l 128-132.

Several reports have claimed that metformin can reverse the bleomycin-induced mouse lung fibrosis pattern [35,37]. However, it is not known whether metformin attenuates TGF-β1-induced pulmonary fibrosis via inhibition of HPMSC transdifferentiation. this mechanism in this cell model has not yet been clarified. Further studies on its mechanism of action and its clinical efficacy are highly warranted.

(35).    Rangarajan, S.; Bone, N.B.; Zmijewska, A.A.; Jiang, S.; Park, D.W.; Bernard, K.; Locy, M.L.; Ravi, S.; Deshane, J.; Mannon, R.B.; et al. Metformin Reverses Established Lung Fibrosis in a Bleomycin Model. Nat Med 2018, 24, 1121–1127, doi:10.1038/s41591-018-0087-6.

(37).    Choi, S.M.; Jang, A.-H.; Kim, H.; Lee, K.H.; Kim, Y.W. Metformin Reduces Bleomycin-Induced Pulmonary Fibrosis in Mice. Journal of Korean Medical Science 2016, 31, 1419–1425, doi:10.3346/jkms.2016.31.9.1419.

  1. The main focus seems to be on TGFb1-induced fibrosis model, but it is not all clear on the use of PDGF.

Thank you for this comment.

We used recombinant TGF-β1 to induce a profibrotic differentiating phenotype of primary human pulmonary MSC whereas PDGF BB treated MSC cells was used to ascertain the effects on MSC cell activation. We commented on the effect of PDGF only when the effects were notable.

  1. Met was used at 5mM for 72h. It would be nice to have a concentration and time response effects.

Thank you for this comment.

We did kinetics with TGF-β1, because we wanted to have fibrosis initiation. This was more marked after 72 hours of treatment. The established fibrosis we treated the cells with metformin. We therefore did not perform kinetics for the metformin treatment, because fibrosis was only identified at 72h.

For the choice of the concentration, we referred to the literature. The aim here was to demonstrate the positive effect of metformin. And to use it as a positive control for our future experiments (1).

(1).  Kheirollahi V, Wasnick RM, Biasin V, Vazquez-Armendariz AI, Chu X, Moiseenko A, et al. Metformin induces lipogenic differentiation in myofibroblasts to reverse lung fibrosis. Nat Commun. 5 juill 2019;10(1):2987.

  1. It appears multiple comparisons were tested against controls only, followed by post hoc analysis of different treatment groups (i.e., TGFb vs TGFb+Met). Are the P values adjusted for multiple comparisons?

Thank you for this comment.

Indeed, the P values are adjusted for multiple comparisons.

  1. Figure 3 shows the effects of PDGF and PDGF+Met, but the results are not described in the main text. Also, the corresponding bar graphs are missing. It appears that Met decreased Col1 expression, as did for PDGF. But, PDGF+Met seems to increase Col1. Please comment on this.

Thank you for this comment.

PDGF and PDGF + met results (fig3) were commented.

“The same effects were observed when we tested the expression of collagen 1 at the protein level (Fig. 3)”.

Referring to

“Treatment with TGF-β1 but not PDGF induced an increase in the expression of all differentiation markers in fibroblasts (Fig. 2A-D),”.

The graphs in Figure 4 have been modified.

  1. Need to provide more details on the pro- or anti-fibrotic miRNAs (miR199, miR214, and miR21) and their specific or reported target mRNAs for post-transcriptional gene regulation. Again, what is the effect of PDGF on these miRs?  Would anti-miRs sufficient to inhibit the action of TGFb?

Thank you for this very accurate comment.

An additional paragraph has been added to the introduction l 82-97.

We did not study the effects of PDGF on miRs, because PDGF did not significantly affect the expression of genes involved in pulmonary fibrosis.

  1. 7. Figure 8. It appears PDGF is equally strong activator of SMAD signaling as TGFb, yet the effects of PDGF on all indices measured related to fibrosis are mild/low. Please comment.

Thank you for this comment.

PDGF is not an activator of the SMAD pathway. In figure 8 there is no difference with the control.

  1. Met is an activator of AMPK. Figure 12B is antithetic to its known role.

Thank you for this very accurate comment.

Activation of the AMPK pathway by metformin has been studied in tissues such as liver, breast cancer, muscle and fat in relation to glucose homeostasis and insulin action, relatively little is known about the effects of this compound on lung mesenchymal cells (HPMSC). Our study is novel in this aspect. Wang et al. showed that cells treated with metformin did not show overexpression of AMPK but increased expression of p-AMPK (3). A study on p-AMPK expression could be performed for future analysis.

(3).      Wang L, Tian Y, Shang Z, Zhang B, Hua X, Yuan X. Metformin attenuates the epithelial-mesenchymal transition of lens epithelial cells through the AMPK/TGF-β/Smad2/3 signalling pathway. Experimental Eye Research. 1 nov 2021;212:108763.

  1. Most of the markers were examined using RT-PCR. How about the protein levels. On that note, does Met have an effect on Nrf2 activity?

Thank you for this comment

We investigated the protein levels for Col1A1, SMA, Ki67 (Fig4), and CXCL12 (Fig 10). Nrf2 was analyzed by PCR. In the laboratory we do not have reagents to perform protein assays on the other markers studied in PCR.

  1. HPMSC was used throughout. How about fibroblasts or myofibroblasts?

Thank you for this comment

The aim of this study was to understand the fate of lung mesenchymal stem cells (HPMSC) following treatment with TGF-β1 (fibrosis inducer), PDGF (cell growth activator) and metformin (potential antifibrotic agent). Thus, we did not study the effects on fibroblasts and myofibroblasts. Other studies very interesting on these cells could be carried out at a later stage but this was not the aim of this work.

The manuscript has been checked and corrected. Thanks for thorough review of this manuscript.

Round 2

Reviewer 1 Report

Well looking into some merits of this manuscript, I gave fair comments. 

Although the authors tried answering the questions, however, seems that the authors diligently or smartly skipped in answering my major questions: in-vivo or ex-vivo data. 

The literature supports some in-vivo models of lung fibrosis. Authors should explore options for generating some data from in-vivo experiments. 

Author Response

Thank you very much for your answer.
We have taken your comments into account.
We have just completed an in vivo pre-study to establish pulmonary fibrosis by CCL4 in C57BL6. The model is therefore established in our laboratory. We are writing the new referral to support our in vitro data. We should have data by the end of 2023.
So we did mention in the conclusion that additional in vitro and in vivo experiments are needed.

"The effects of Met promoting transdifferentiation (process also called fibrosis reversion) of myofibroblasts towards lipofibroblasts and osteoblasts is promising and further experiments along these lines are now highly warranted in vitro but also benefiting from an animal model of lung fibrosis. "

l589

We would like to thank reviewer 1 for his comments and corrections on our manuscript.

Reviewer 2 Report

no further comments.

Author Response

Thank you for taking the time to review our article.